



# A deeper look at the relationship between root carbon pools and the vertical distribution of the soil carbon pool

Ranae Dietzel[1], Matt Liebman[1], Sotirios Archontoulis[1]

[1]Department of Agronomy, Iowa State University, Ames, 50011, United States

*Correspondence to*: Ranae Dietzel (ranae.dietzel@gmail.com)

**Abstract.** Plant root material makes a substantial contribution to the soil organic carbon (C) pool, but this contribution is disproportionate below 20 cm, where 30% of root mass and 50% of soil organic C is found. Root carbon inputs changed drastically when native perennial plant systems were shifted to cultivated annual plant systems. We used the reconstruction of a native prairie and a continuous maize field to examine both the relationship between root carbon and soil carbon and the fundamental rooting system differences between the

vegetation under which the soils developed versus the vegetation under which the soils continue to change. In all treatments we found that root C:N ratios increased with depth, which may help explain why an unexpectedly large proportion of soil organic C is found below 20 cm. Measured root C:N ratios and turnover times along with modelled root turnover dynamics showed that in moving from prairie to maize, a large, structural-tissue dominated root C pool with slow turnover, concentrated at shallow depths was replaced by a small, non-structural-tissue dominated root C pool with fast turnover evenly distributed in the soil profile. These differences in rooting

systems suggest that while prairie roots contribute more C to the soil than maize at shallow depths, maize may contribute more C to the soil than prairies at deeper depths.

## 1 Introduction

Prairie-formed Mollisols support some of the world's most productive agriculture, but declines in levels of soil organic matter threaten the reliability of this production. Soil organic matter losses coincide with a shift from perennial plant systems to annual cropping systems

that introduced frequent tillage, subsurface drainage, and differences in organic matter inputs, including considerably different rooting systems (Davidson and Ackerman, 1993; Huggins et al. 1998; Guo and Gifford, 2002). The effects of changes in aboveground processes such as increased soil disturbance and aeration, addition of fertilizers, and changes in residue amount and quality have often been cited as primary factors in the changes of soil organic matter from native levels (Buyanovsky et al. 1987, Huggins et al. 1998, David et al. 2009, Gregory et al. 2016). The role played by changes in rooting systems, on the other hand, is difficult to study and has received less

attention.

In this paper, we distinguish between a root C pool defined as C found in any material that can still be visually identified as a root and a soil organic C pool defined as the rest of the soil organic C. Root growth allows the placement of plant tissue directly into the soil, creating a root C pool as deep as the rooting system occupies. Some studies suggest that root C pool size and soil organic C pool size have a direct relationship and that most soil organic matter is derived from roots (Balesdent and Balabane 1996, Rasse et al. 2005, Kong

and Six 2010). This would mean that a change in root inputs, such as that engendered by switching from annual to perennial systems,

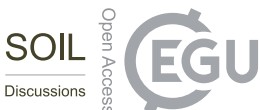

would have a direct impact on soil organic matter even deep into the soil profile. However, few direct comparisons of annual and perennial rooting systems have been made and our understanding of soil C dynamics decreases as soil depth increases.

On average, half of the world's soil C is found below 20 cm (Rumpel and Kogel-Knabner, 2011). However, only 30% of the world's roots are found below 20 cm (Jobbagy and Jackson, 2000). In the central US, this phenomenon was observed as early as 1935 when

Weaver found 41-74% of the total soil organic matter, but only 23-29% of the total root mass in a tallgrass prairie were below 20 cm. Similarly, Gill and colleagues (1999) found 77% of total soil organic matter, but only 43% of total root mass below 15 cm in a shortgrass steppe. Although this disproportionate relationship between root and soil C distribution has been known for some time, no widely accepted explanation exists to explain the magnitude of difference between the amount of C in the root pool and the amount of C in the soil pool (Gill et al. 1999, Rumpel and Kogel-Knabner 2011).

Many factors interact to determine how much C is transferred between pools and how much C remains in a particular pool. Soil temperature, moisture, and $O_2$ availability are the most important environmental variables controlling the rate of decomposition (Gill and Burke, 2002) and soil texture and existing soil C levels determine the length of time C remains in the soil (Six et al. 2002; Rasse et al. 2005). The C:N ratio of the organic matter being decomposed also plays a key role in both the rate of decomposition and the fate of the decomposed organic matter, with higher C:N ratios leading to slower decomposition (Silver and Miya 2001) and fewer microbial

by-products (Cotrufo 2015). Temperature, moisture, $O_2$, soil texture, and soil C levels all vary with soil depth and contribute to partial explanations of the size discrepancy between root and soil C pools. However, previous studies that measured roots and/or organic matter with depth have neglected to report the change of root C:N ratio with depth (Tufekcioglu et al. 2003, Beniston et al. 2014). Carbon:N ratio differences between maize and prairie root C pools are also unknown. A more-detailed look at properties of root C pools is needed.

We examined the belowground reconstruction of native vegetation on a Mollisol after >100 years of annual cropping to gain a unique

perspective on characteristics of root inputs that would not necessarily be detected in established prairie systems, but that contribute to dynamics of the belowground ecosystem. We examined differences between maize and reconstructed prairie root pools to a depth of one meter to serve two separate, but related, purposes: 1) inform our understanding of the impacts of shifting millions of hectares from perennial to annual vegetation, and 2) contribute to an explanation of why levels of soil organic C found below 20 cm are greater than expected based on root distribution. In comparing the root C pool of a reconstructed prairie system to the root C pool of a maize cropping

system we asked the following questions: 1) how does the quantity, distribution, and C:N ratio of the root C pool differ with depth and between these native perennial and non-native annual ecosystems? 2) what do these differences in inputs tell us about the historical belowground ecosystem under which these soils developed and the systems under which these soils continue to change?

## 2 Materials and Methods

### 2.1 Site Conditions and Experimental Design

We conducted the experiment in Boone County, IA, USA on the Iowa State University Agronomy and Agricultural Engineering Research Farm (41°55′N, 93°45′W). Soils at the site were primarily Webster silty clay loam (fine-loamy, mixed, superactive, mesic Typic Endoaquolls) and Nicollet loam (fine-loamy, mixed, superactive, mesic Aquic Hapludolls). The 60-year mean growing season precipitation 11 km from the site was 720 mm. Prior to initiation of the field experiment in 2008, the site was used for maize and soybean production and was planted with soybean in 2007. Soil sampling to 15 cm in November 2007 indicated mean soil pH was 6.7,





mean soil C concentration (via dry combustion analysis) was 30 g kg$^{-1}$, mean extractable phosphorus concentration (via Bray-1 procedure) was 11 mg kg-1, and mean extractable potassium (via Mehlich-3 procedure) was 141 mg kg$^{-1}$.

Experimental plots were 27 m x 61 m and were arranged as a spatially balanced complete block design (van Es et al., 2007). The three cropping systems used for the present study were continuous maize with annual removal of grain and about 50% of the stover (hereafter

maize), reconstructed multispecies prairie with annual aboveground biomass removal (hereafter unfertilized prairie), and N-fertilized reconstructed multispecies prairie with annual aboveground biomass removal (hereafter fertilized prairie). All of the treatments were managed without tillage. Conventional farm machinery was used for planting, fertilization, crop protection, and harvest operations. Herbicides were not used in the prairie systems except for a small number of spot treatments for Canada thistle (*Circium canadense*) control, and the timing and frequency of herbicide use in the annual cropping systems varied among treatments.

Both prairie treatments were sown on 19 May 2008 with the same custom seed mix obtained from Prairie Moon Nursery (Winona, MN, USA) that contained 31 species, including C3 and C4 grasses and leguminous and non-leguminous forbs. All species were perennial and sourced from within 240 km of the experiment site. The composition of the seed mix by weight was 12% C3 grasses, 56% C4 grasses, 8% legumes, and 24% non-leguminous forbs. A detailed description of the prairie plant community compositions can be found in Jarchow and Liebman (2013). The fertilized prairie treatment received no fertilizer in 2008 (the establishment year), but was fertilized

at a rate of 84 kg N ha$^{-1}$ year$^{-1}$ in all subsequent years. This fertilizer rate was chosen because it was similar to the maximum rate of pre-planting N fertilization recommended for maize (Blackmer et al., 1997) and the expected N removal in the harvested biomass of perennial grasses grown in the area (Heggenstaller et al., 2009).

The maize used was a 104-day relative maturity hybrid (Agrigold 6325 VT3) with transgenes for glyphosate resistance, corn borer (*Ostrinia nubilalis*) resistance, and corn rootworm (*Diabrotica* spp.) protection. Maize was planted following standard practices

(Abendroth et al., 2011) in rows spaced 76 cm apart at a seeding rate between 79,500 seeds ha$^{-1}$ and 82,500 seeds ha$^{-1}$, depending on the year. Fertilizer rates for corn were based on soil testing results (Blackmer et al. 1997) and varied from 123 to 200 kg N ha$^{-1}$, depending on the year.

### 2.2 Data Collection

#### 2.2.1 Soil Collection

Soil cores were taken to 1 m depth in all plots each year using a hydraulic soil probe (Giddings Machine Co., Windsor, CO, USA) after all crop and prairie plots were harvested. Sampling occurred by replicate block from 31 October-25 November 2008, 9-11 November 2009, 25-28 October 2010, 28-31 October 2011, 16-17 October 2012, and 7-11 October 2013.

In 2008, two cores were taken per plot. A 0-30 cm fraction was taken with a 10.2 cm internal diameter soil probe; the 30-100 cm fractions of the cores were taken within the same hole as the 0-30 cm fraction, but with a smaller soil probe. In Blocks 1 and 4, the

internal diameter of the core was 6.0 cm. In Blocks 2 and 3, the internal diameter of the core was 5.2 cm. In 2009 and 2010, four cores were taken per plot. The 0-30 cm fraction of the cores was taken with a 10.2 cm internal diameter soil probe; the 30-100 cm fractions





of the cores were taken directly below the 0-30 cm fraction with a 5.1 cm internal diameter probe.  In 2011-2013, four cores were taken per plot, and the entire core was taken with a 5.1 cm internal diameter probe.

Soil cores were ultimately divided into three or five depth increments.  In 2008, depth increments were 0-30 cm, 30-60 cm, and 60-100 cm.  In 2009-2013 depth increments were 0-5 cm, 5-15 cm, 15-30, cm, 30-60 cm, and 60-100 cm.  Following division and extraction

from the field, soil cores were stored at 5°C until processing was initiated.

Each year, 60-100 g of root-free soil was removed from each depth increment, air-dried, and archived in airtight containers at room temperature. In 2008 and 2013, this soil was ground on a roller-mill and organic C content was determined by catalytic oxidation and $CO_2$ measurement with NDIR in an Elementar TOC Cube at Brookside Laboratories, Inc. (New Bremen, OH, USA).

**2.2.2 Root Pool Collection**

Two sets of root pool measurements were collected: a) end-of-season root data for each year (depth 0 to 100 cm) and b) in-season root measurements during 2010 and 2011 (depth 0-30 cm). The first, described in this section, was used to track changes in the root C pools over all six years and the second, described in section 2.2.3, was used to quantify annual root C contributions in 2010 and 2011.

Root extraction from the soil began by washing the soil samples described in 2.2.1 in wire mesh tubes (0.28 mm mesh) for 3 h in an

elutriator (Wiles et al., 1996).  Roots were removed from the remaining soil by suspending the air-dried sample in water and collecting the roots, which floated, with sieves followed by manually removing any remaining non-root material that was present in the samples. Any plant crowns that were present in the samples were removed and were not considered to be root biomass.  Roots were then dried at 70°C for at least 4 h before being weighed.  All root samples were ground to 2 mm with a centrifugal mill and concentrations of C and N were determined by combustion analysis at the Soil and Plant Analysis Laboratory at Iowa State University (Ames, IA, USA).

**2.2.3 In-season Root Growth**

In 2010 and 2011, root biomass was measured with an in-situ growth core approach (Neill 1992) to capture only those roots growing within the measurement year.  After fall harvest in 2009 and 2010, eight 10.2-cm-diam soil cores were taken to 30 cm depth in each plot and brought to the laboratory.  Holes created in the field were held open during the winter by capped 10.2 cm PVC piping.  In the laboratory, cores were divided into 10 cm sections and virtually all roots were removed by hand.  Soil was stored in intact cores at 30°C

for the first year of the experiment (intended to be used for an incubation experiment) and 4°C in sealed plastic bags for second year of the experiment.  The differences in storage conditions did not have an apparent effect on the outcome of the experiment.  At the end of winter while plants were still dormant, the root-free soil was returned to its original location in the field in 10 cm depth increments. Soil was packed to imitate the surrounding bulk density, approximately 1.4 g cm$^{-3}$.  Root-free zones were located randomly within prairie plots and at 20 cm from maize rows.  Eight root-free areas were situated within each plot, allowing duplicate sampling at four time

points throughout the growing season.  Two 4-cm-diameter soil cores were taken within each 10.2-cm-diam root-free area to a 30 cm



depth at each root sampling date. Bulk soil was washed from the roots with water using a soil elutriator (Wiles et al., 1996), roots were dried at 60° C for 24 hours, non-root biomass was removed from the roots by hand, and roots were weighed.

**2.3 Data Analysis**

Root pool mass for the entire meter depth was calculated by summing together the root mass for each depth increment of an entire core and whole core root masses were compared between treatments within each year using contrasts within a linear mixed effect model in the R package nlme (Pinheiro et al. 2013). Treatment differences within depths within years and differences between treatments within depths within years for root biomass were also made with contrasts with linear mixed effects models using proc glimmix in SAS (SAS Institute, 2011).

Because root mass in 2008 was measured at three increments (0-30 cm, 30-60 cm, and 60-100 cm) instead of the five increments used later in the experiment (0-5 cm, 5-15 cm, 15-30 cm, 30-60 cm, and 60-100 cm), 2008 root mass for 0-5 cm, 5-15 cm, and 15-30 cm depths was estimated by multiplying the average 2009-2013 depth distribution proportions by the 2008 0-30 cm increment. No important comparisons were made using this estimated data, but the data were used as a starting point for graphing C:N ratios in different depth increments and fitting curves to root accumulation. Carbon:N ratios were compared between treatments within years within depths and between years within treatments within depths using proc glimmix in SAS.

Root mass measured at the end of each growing season was subset by depth increment and each subset was fit by both a logistic model and a linear model for each plot. Logistic models and linear models were compared against each other using Akaike's Information Criterion (AIC) and the model with the lowest AIC was chosen. The AIC was not greatly different for any of the comparisons, but the logistic model had the best fit for every depth. Model fits and comparisons were done using the R package nlme (Pinheiro et al. 2013).

The first derivative of the logistic model was used to calculate the daily rate of root mass accumulation. Parameters from the logistic model were used to predict both amount and rate of accumulation for each day for each depth in each plot of the experiment. These predictions were averaged for each treatment and plotted. The annual mean rate was calculated by averaging accumulation rates across each growing season for each depth in each plot. Comparisons of rates between treatments within depths and within years and comparisons of rates between depths within treatments within years were made with proc glmmix in SAS.

Models used to fit root mass over time did not accurately reflect within year biomass fluctuations caused by the start and stop of plant growth and freezing and thawing of soil, rather these curves were used to compare long-term trends in root mass accumulation. Accordingly, the daily rate of root mass accumulation was also inaccurate, but very useful to compare relative accumulation rates among treatments and soil depths. An average daily root mass accumulation rate was calculated by considering the period of possible root growth and decomposition to be between April 1st and November 30th of each year.

In-situ root measurements in 2010 and 2011 combined with differences in root pool masses at 30 cm over these years were used to calculate a root turnover constant ($k$) and root mean residence time (mrt) using the equations $k = loss/pool$ and $mrt = 1/k$. Root pool




loss during each year was calculated as the difference between the mass accumulated during that year and the gain measured by in-situ growth cores. The root mass measured at the end of each year was the pool value.

The height and volume of root samples varied among depth increments, making visual comparisons among depths, such as 0-5 cm and 60-100 cm, difficult. Thus, splines were fit to the data and integrated by 5 cm depths to facilitate visualization of root and soil organic
5    C distribution in the soil profile.

### 3 Results

Table 1. Soil characteristics measured at the establishment of the experiment.

| Depth | Bulk Density (g cm⁻³) | pH | Total C (%) | Total N (%) | Sand (%) | Silt (%) | Clay (%) |
|---|---|---|---|---|---|---|---|
| 0-5 | 1.28 | 6.36 | 2.81 | 0.24 | 37.5 | 36.8 | 25.8 |
| 5-15 | 1.41 | 5.85 | 2.55 | 0.22 | 37.5 | 36.0 | 26.6 |
| 15-30 | 1.50 | 5.94 | 2.14 | 0.18 | 35.4 | 35.8 | 28.9 |
| 30-60 | 1.45 | NA | 1.23 | 0.11 | NA | NA | NA |
| 60-100 | 1.60 | NA | 0.95 | 0.05 | NA | NA | NA |

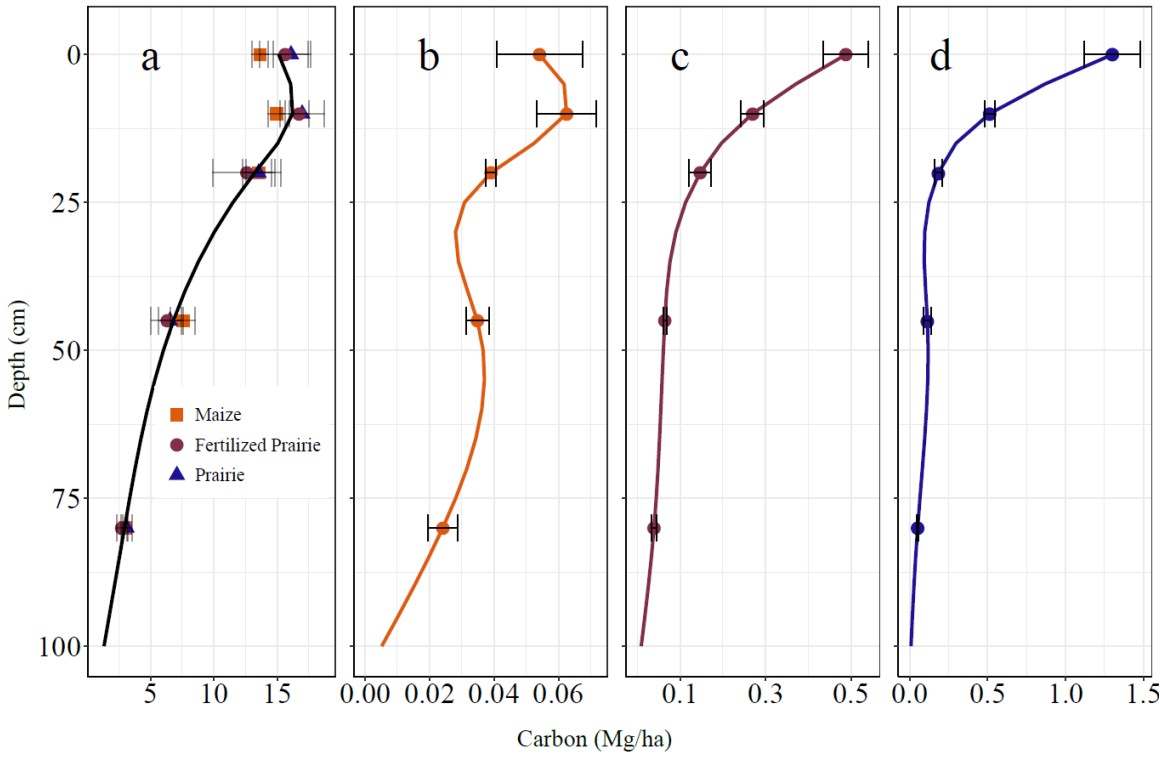

Figure 1. a) Total soil carbon with each treatment represented by a point and the site average represented by a solid line, b) maize root carbon, c) fertilized prairie root carbon, d) unfertilized prairie root carbon measured in 2013, six years after establishment of the experiment. Different x-axes scales are used to emphasize similarities and differences in profile distribution patterns, not absolute mass amounts (see Fig 2).

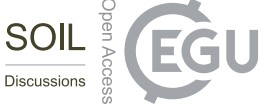



The total amount of organic C found in the soil 6 years after establishment of the experiment did not differ among treatments at any depth (Fig 1), nor was it different from initial total organic C levels measured at the beginning of the experiment (data not shown). Half of the total soil organic C was found below 20 cm (Table 2). The pattern of vertical soil C distribution was similar to the pattern of maize root distribution, not prairie root distribution (Fig 1).

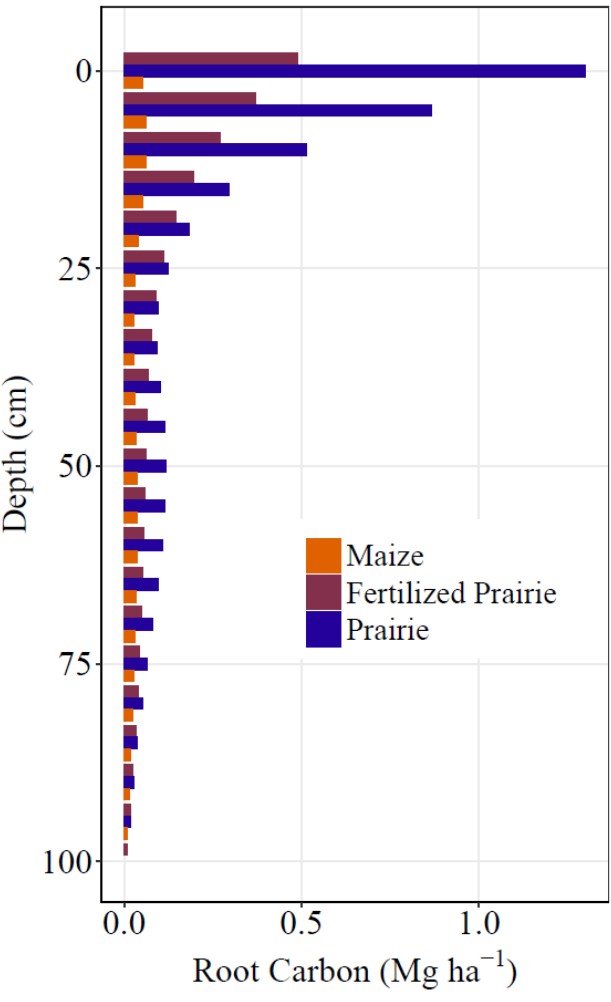

Figure 2. Absolute difference in root C pools six years after prairie establishment.

Six years after the establishment of the experiment, the unfertilized prairie root C pool was almost 6 times greater than the maize root
10  C pool and the fertilized prairie root C pool was 3.5 times greater than the maize root C pool over a 1 m depth. Twenty-eight percent of the unfertilized prairie root C pool, 37% of the fertilized prairie root C pool and 62% of the maize root C pool was found below 20 cm (Fig 2, Table 2).





Table 2. Root pool and soil organic C found above and below 20 cm.

| Treatment | Depth (cm) | Root C (Mg) | Soil C (Mg) | Root C (proportion) | Soil C (proportion) |
|---|---|---|---|---|---|
| Maize | 0-20 | 0.27 | 71.17 | 0.38 | 0.44 |
| | 20-100 | 0.43 | 89.97 | 0.62 | 0.56 |
| Unfertilized Prairie | 0-20 | 3.16 | 79.14 | 0.72 | 0.48 |
| | 20-100 | 1.26 | 85.00 | 0.28 | 0.52 |
| Fertilized Prairie | 0-20 | 1.47 | 76.66 | 0.63 | 0.50 |
| | 20-100 | 0.85 | 76.54 | 0.37 | 0.50 |

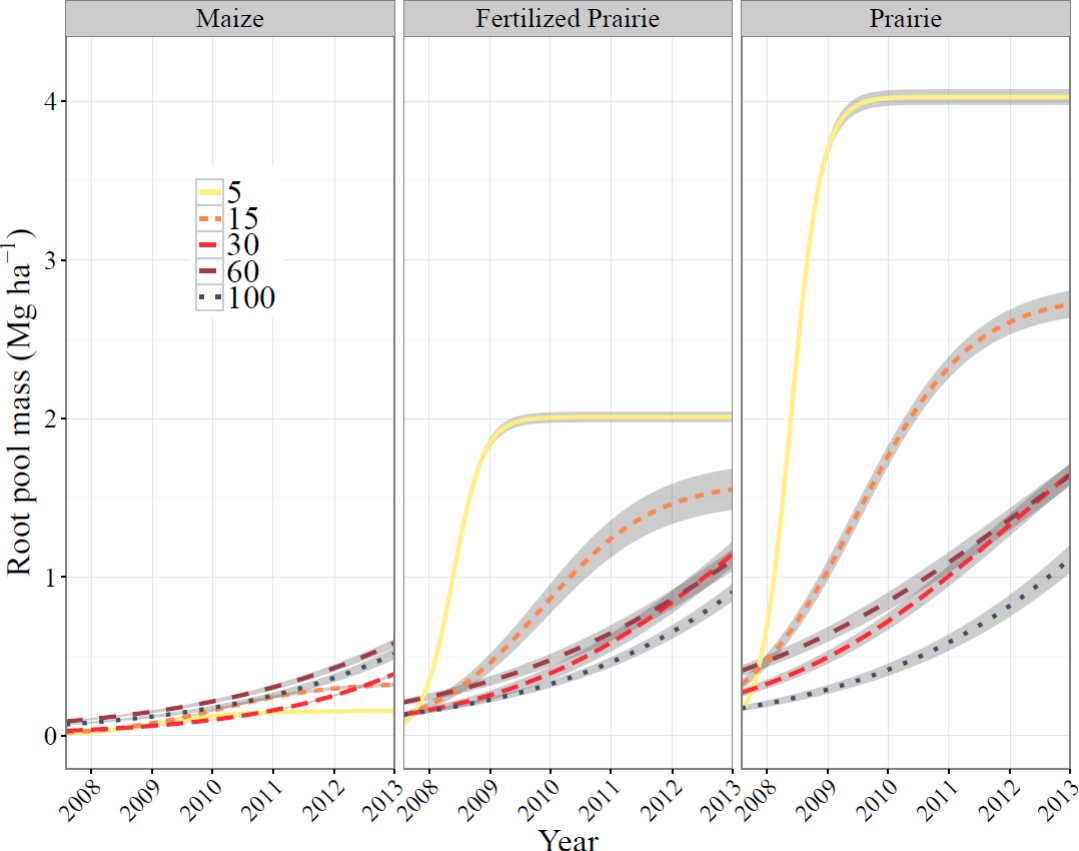

5     Figure 3. Modeled accumulation of root pool mass over six years at 0-5 cm, 5-15 cm, 15-30 cm, and 30-60 cm, and 60-100 cm. Grey shading represents one standard error of the mean. Seasonal effects are smoothed.



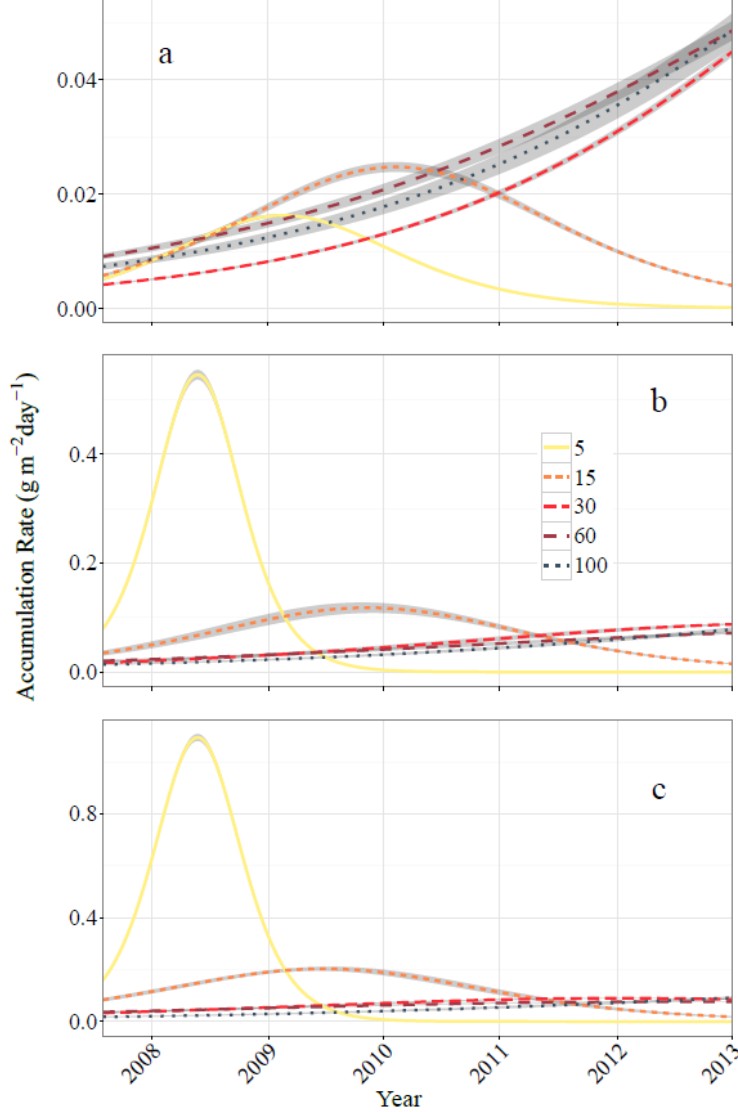

Figure 4. Modeled rates of root pool mass accumulation over 6 years in a) continuous maize, b) fertilized prairie and c) unfertilized prairie at 0-5 cm, 5-15 cm, 15-30 cm, 30-60 cm, and 60-100 cm. Grey shading represents one standard error of the mean. Different y-axes are used to emphasize similarities and differences in timing as well as to make within treatment relationships easier to see. Seasonal effects are smoothed.



Table 3. Root pool accumulation rates averaged across each growing season (g m$^{-2}$ day$^{-1}$). Differences in lowercase letters indicate significant differences between depths within treatments within years (read up and down). Differences in uppercase letters indicate differences between treatments within depths within years (read left to right).

| Year | Depth (cm) | Maize (g m$^{-2}$ day$^{-1}$) | | | Fertilized Prairie (g m$^{-2}$ day$^{-1}$) | | | Unfertilized Prairie (g m$^{-2}$ day$^{-1}$) | | |
|---|---|---|---|---|---|---|---|---|---|---|
| 2008 | 0-5 | 0.007 | a | C | 0.205 | a | B | 0.411 | a | A |
| | 5-15 | 0.007 | a | C | 0.044 | b | B | 0.102 | b | A |
| | 15-30 | 0.005 | a | C | 0.019 | c | B | 0.036 | c | A |
| | 30-60 | 0.010 | a | C | 0.025 | c | B | 0.058 | a | A |
| | 60-100 | 0.008 | a | B | 0.015 | c | AB | 0.019 | a | A |
| 2009 | 0-5 | 0.015 | a | C | 0.315 | a | B | 0.632 | a | A |
| | 5-15 | 0.016 | a | C | 0.087 | b | B | 0.177 | b | A |
| | 15-30 | 0.007 | a | C | 0.029 | c | B | 0.051 | c | A |
| | 30-60 | 0.015 | a | C | 0.036 | c | B | 0.084 | d | A |
| | 60-100 | 0.012 | a | B | 0.021 | c | AB | 0.027 | e | A |
| 2010 | 0-5 | 0.013 | a | A | 0.011 | d | AB | 0.021 | d | A |
| | 5-15 | 0.024 | a | D | 0.117 | a | B | 0.197 | a | A |
| | 15-30 | 0.012 | a | C | 0.042 | bc | B | 0.067 | c | A |
| | 30-60 | 0.020 | a | C | 0.047 | b | B | 0.090 | b | A |
| | 60-100 | 0.016 | a | BC | 0.030 | c | AB | 0.037 | d | A |
| 2011 | 0-5 | 0.005 | a | A | 0.000 | c | AB | 0.000 | e | A |
| | 5-15 | 0.022 | a | D | 0.093 | a | B | 0.131 | a | A |
| | 15-30 | 0.018 | a | C | 0.058 | b | B | 0.082 | b | A |
| | 30-60 | 0.027 | a | C | 0.056 | b | B | 0.068 | c | A |
| | 60-100 | 0.023 | a | C | 0.041 | b | AB | 0.051 | d | A |
| 2012 | 0-5 | 0.001 | c | A | 0.000 | c | A | 0.000 | d | A |
| | 5-15 | 0.012 | b | D | 0.048 | b | B | 0.061 | b | A |
| | 15-30 | 0.028 | a | D | 0.074 | a | B | 0.089 | a | A |
| | 30-60 | 0.034 | a | B | 0.058 | b | A | 0.041 | c | B |
| | 60-100 | 0.033 | a | D | 0.056 | b | B | 0.068 | b | A |
| 2013 | 0-5 | 0.000 | b | A | 0.000 | e | A | 0.000 | c | A |
| | 5-15 | 0.005 | b | B | 0.019 | d | A | 0.023 | b | A |
| | 15-30 | 0.041 | a | D | 0.086 | a | A | 0.087 | a | A |
| | 30-60 | 0.041 | a | B | 0.052 | c | A | 0.022 | b | C |
| | 60-100 | 0.045 | a | C | 0.074 | b | B | 0.087 | a | A |




Prairie rooting systems were established sequentially in the soil profile from the top down. The top five cm of the root pool peaked in the first full year of growth and then reached an equilibrium during the second full year of growth with large year-to-year variability given the sensitivity of this thin surface layer to environmental conditions (Fig A1-A3). The next soil layer, from 5-15 cm, had the greatest increase in root pool mass during the second full year of prairie growth, whereas, in contrast, the 15-30 cm and 30-60 cm depths

didn't reach peak rates of root pool accumulation until five and six years after establishment, with no indication of when accumulation would cease. In the unfertilized prairie, rates of root pool accumulation in the 60-100 cm of the soil in the sixth year were greater than all other depths with no sign of slowing down. Fertilized prairie also had a high rate of root pool accumulation at 60-100 cm in the sixth year with no sign of decreasing.

Maize root pool accumulation was almost always slower than prairie root pool accumulation with the exception of the top 5 cm after 2010, 60-100 cm before 2011 (not different from fertilized prairie), and a greater value in maize than unfertilized prairie at 30-60 cm in 2013. There was no difference in maize root pool accumulation among depths until 2011 when accumulation below 15 cm then began to exceed accumulation above 15 cm.

Table 4. Root turnover at 0-30 cm.

| Year | Treatment | Input (g $m^{-2}$) | Gain (g $m^{-2}$) | Loss (g $m^{-2}$) | Pool (g $m^{-2}$) | k | mrt (years) |
|------|-----------|------|------|------|------|------|------|
|      | Unfertilized Prairie | 367 | 104 | 263 | 748 | 0.35 | 2.85 |
| 2010 | Fertilized Prairie | 146 | 62 | 84 | 231 | 0.37 | 2.74 |
|      | Maize | 56 | 18 | 38 | 44 | 0.86 | 1.16 |
|      | Unfertilized Prairie | 387 | 78 | 309 | 758 | 0.41 | 2.45 |
| 2011 | Fertilized Prairie | 168 | 55 | 113 | 342 | 0.33 | 3.02 |
|      | Maize | 48 | 16 | 31 | 47 | 0.67 | 1.50 |

Prairie roots had a mean residence time (mrt) of 2.75 years in the top 30 cm of the profile when averaged across treatments and years (2010 and 2011). Maize roots turned over twice as fast as prairie roots when averaged across treatments and years (Table 3).

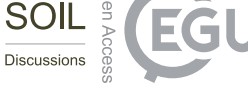


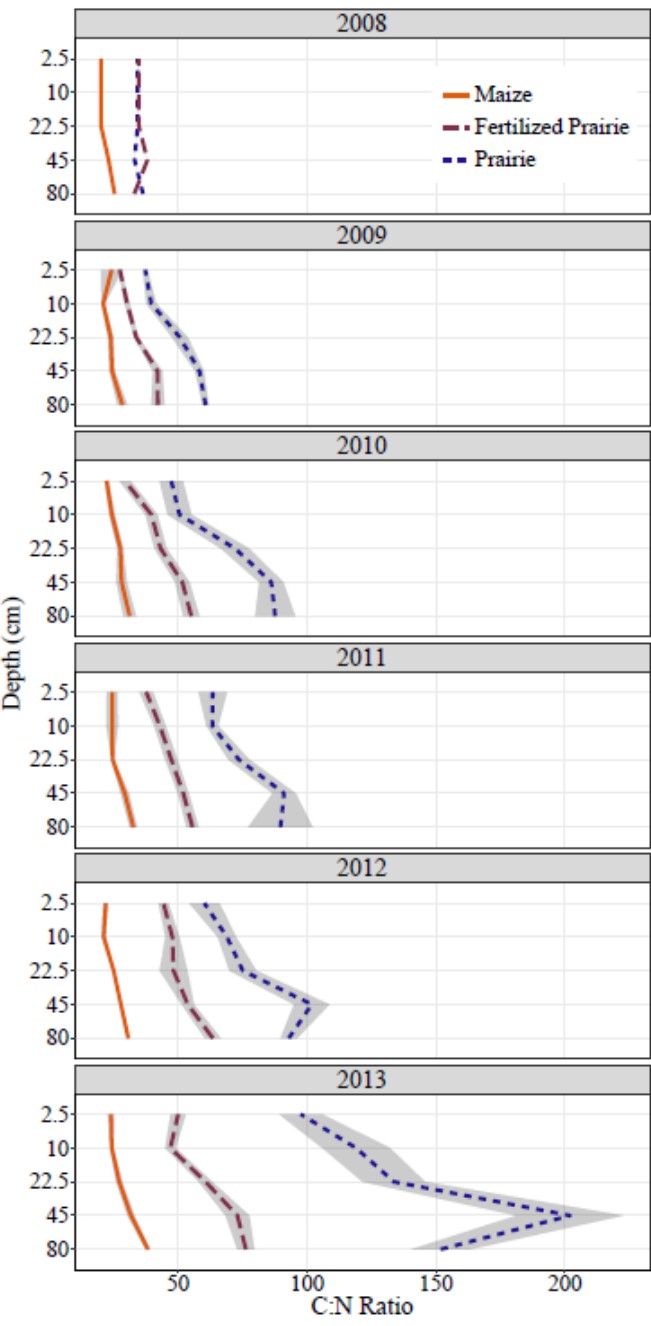

Figure 4. Root C:N ratios with depth over time. Grey shading represents one standard error of the mean.

Carbon:N ratios increased with depth in all treatments. Carbon to nitrogen ratios increased in both prairie treatments in every depth
5 over time, although the increase in fertilized prairie was not always different between consecutive years. In all treatments, changes in
C:N ratios were the result of both an increase in C content and a decrease in N content (data not shown). The maize root pool did not
exhibit an increase in C:N ratio over time.



## 4 Discussion

### 4.1 Reconstruction of a prairie root C pool and implications for C contribution

An increase in root pool C:N ratio with depth has not been reported previously in the literature and not previously considered when trying to determine why a disproportionately large amount of soil C is found at depth when compared to root distribution. It has been

recently theorized that plant tissue becomes organic matter through two different pathways: 1) a dissolved organic C-microbial pathway whereby plant litter is first processed by soil microbes and eventually transported and stabilized in the soil matrix as a microbial by-product, if the soil has the capacity to stabilize these compounds, and 2) a physical-transfer pathway whereby plant tissue is processed by soil microbes to its fullest extent, and then remains in the soil functionally inert (Cotrufo 2015). In the present study, the former pathway is more applicable to tissue dominated by non-structural compounds, such as that with lower C:N ratios found here at shallower

depths, whereas the latter pathway applies to tissue dominated by structural compounds, indicated by high C:N ratios in root tissue at greater depth. Under this framework, root decomposition in our study would have resulted in a gradient of microbially-derived to physically-derived organic matter from the top of the soil profile downward. In turn, this would mean that soil organic matter at the soil surface would be vulnerable to transport to greater depth as dissolved organic C, whereas physically-transferred soil organic matter at depth would be relatively immobile. This is a possible mechanism by which the amount of soil organic C found at depth is

disproportionately large compared to the size of the root C pool. These findings are consistent with evidence that the contribution of microbial- and not root-derived C increases with depth (Liang and Balser 2008, Rumpel and Kogel-Knabner, 2011). In addition to the less-structural root material found at shallow depths, these areas of concentrated roots produce labile exudates that are easily metabolized and transported deeper in the soil profile (Badri and Vivanco, 2009). While we did not measure root exudates, it is important to recognize that these mobile compounds also likely play an important role in the development of the soil organic C profile.

Because the root pool is made up of a combination of new, mature, ageing, and dead roots, an increase in its mass comes from root growth, live root retention, and inhibited root decomposition. The relatively quick rate of accumulation in the top 30 cm of soil was most likely a function of new root addition, which slowed as the system became more established. Slower increases at deeper depths than shallower depths may indicate that accumulation there is more dependent upon the carryover of roots from previous years than at

shallower depths.

By the sixth year of reconstructed prairie establishment, root C pool equilibrium was reached and prairies began making substantial annual contributions to the soil organic matter pool above 30 cm, although the fraction of organic matter that remained in the soil is unknown. This was indicated by the finding that the majority of prairie roots (75%) was found in this depth fraction, where mean

residence time was measured to be 2.5-3 years. The prairie root C pool at 0-5 cm reached an equilibrium and began steady root turnover in the third year after establishment, as indicated by very low rates of accumulation, and was likely able to contribute material to the soil organic matter pool at this time. Other prairie restorations have also found soil organic matter accumulation to be most rapid closer to the soil surface (O'Brien et al. 2010, Omonode and Vyn 2006).

Annual prairie root inputs were not measured below 30 cm, so turnover rates could not be calculated. However, continuous increases in the root pool at depth due to root growth and retention indicate that root tissue loss to the soil was very low during this time and the mean residence time of roots at depth may have greatly exceeded those closer to the surface. This means that at depth, not only was the root C pool substantially smaller than near the surface, but root material also became available to the soil much more slowly than near the surface. Indeed, DuPont et al. (2014) found intact prairie roots in the soil five years after conversion to annual wheat.






Nitrogen fertilization of prairies led to a smaller root pool at every depth, with lower rates of accumulation, and lower C:N ratios. However, fertilization did not affect the time until root systems were fully established or the turnover rate of roots in the top 30 cm. Differences between fertilized and unfertilized prairie showed that the pattern of distribution was a function of nutrient availability and not a response to soil space conditions because fertilized prairie used half as much space as unfertilized prairie and still showed decreased

accumulation above 30 cm over time.

### 4.2 Quantity, distribution, and quality of root biomass differs in native perennial and non-native annual ecosystems

It is possible that maize roots contributed more C to the soil than did prairie roots below a certain depth. Maize root C pools were much smaller than prairie root C pools, but faster turnover times and lower C:N ratios resulted in a greater proportion of the maize root C pool available for stabilization in the soil compared to the prairie root C pool. In the top 0-30 cm, the difference in mass between even the fertilized prairie and maize was too great to be overcome by faster turnover and greater carbon use efficiency, but the difference in root mass between the annual and perennial systems decreased with depth while the difference in C:N ratio increased and turnover times

may have maintained the same relative relationship.

### 4.3 What do these differences in inputs tell us about the historical belowground ecosystem under which these soils developed in comparison to the systems under which these soils continue to change?

The experimental location was a site of cultivation under annual crops for over 100 years. We do not have measurements of the pre-

cultivation soil C profile, but other data from sites near our experiment (Guzman 2009, McGranahan et al. 2014) show that the soil C profile shifted from a pattern of having an exponential decrease in C with distance from the surface to a pattern of more uniform distribution of C with the highest point of C 10 cm below the surface. The loss of C in the soil surface after cultivation is well known and attributed to mass loss through soil erosion, increased mineralization of organic matter through tillage, and decreased belowground organic matter inputs (Davidson et al. 1993; Huggins et al. 1998). Less is documented about the change in soil carbon below 30 cm,

but using a robust dataset, Veenstra et al. (2015) found soil organic C to increase below 35 cm after 50 years in maize and soybean cropping systems in Iowa. Initial soil organic C measurements in that study were made ~50 years after these soils had already been converted to annual systems, preventing comparison to soil organic C levels at depth under native vegetation, but results from Veenstra et al. still show that Mollisols can and do gain soil organic C at deeper depths under maize and soybean systems. Similarly, David et al. (2008) and Follett et al. (2009) found cultivated sites that gained deep soil organic C relative to remnant prairies and grasslands.


Our relatively short-term study of 6 years did not detect significant changes in soil C at any depth, but differences in quantity, distribution, and C:N ratios between the annual and perennial rooting systems we studied have important implications for how deep soil organic C may have changed and continues to change with the implementation of annual cropping systems. A large, structural-tissue dominated root C pool with slow turnover, concentrated at shallow depths was replaced by a small, non-structural-tissue dominated root

C pool with fast turnover evenly distributed in the soil profile. The difference in size between these two pools has long been obvious, but often misleading for comparisons related to C accounting because differences in root turnover and tissue C:N ratio are often not taken into consideration. An exception, Omonode et al. (2006) discusses the possibility that slower turnover in perennial rooting systems may prevent expected increases in soil organic C compared to adjacent maize systems. The data presented here in the context of recent organic matter formation theory suggest that while differences in root C pool and soil organic C relationships in maize and prairie above

20 cm are predominately controlled by root biomass amount, root biomass amount is less of a factor below 20 cm.



## 5 Conclusion

Soils are incredibly complex systems and biogeochemical processes that determine soil C storage happen over a long time and in places that are difficult to study without artifact-inducing disturbances. We have shown here that an increase in root C:N ratio with depth is a

potentially important, and previously unconsidered, factor determining the distribution of C in the soil profile. This factor interacts with depth-determined differences in soil temperature, moisture, $O_2$, soil texture, microbial communities, and existing soil C content and thus carries different significance in different environments. In our comparison of maize and reconstructed prairie systems, root pool C:N ratios may be sufficiently important that they result in greater maize C contributions to soil organic matter than prairie C contributions to soil organic matter below 20 cm. In these and many other herbaceous systems, an increase in root C:N ratio with an increase in depth

may in part explain why 50% of soil organic C is found below 20 cm, while only 30% of root biomass is found in the same location. Elucidating the mechanisms determining soil C retention and addition is important as we strive to design systems that maintain and build soils that are productive and resilient. The role of roots and root composition, as well as the importance of soil organic C below 20 cm should be carefully considered in such designs.

**Data and code availability**

Data and code for this work is currently publicly stored in a GitHub repository. During and after the review process, we will clean up the repository to include only relevant data and code, improve comments within the code, and write a thorough readme file to ensure the creation of a fully reproducible compendium. The GitHub repository will be linked with a Zenodo account, which will provide a DOI for the data and code, making the data easily discoverable and citable. Zenodo will also create a mirrored repository, backing up

code and data in their own system.





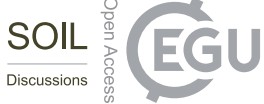



**Appendix A.**

*Curve fits used to generate predicted root accumulation for each depth. The mean and standard error of these curves are found in Fig. 3.*

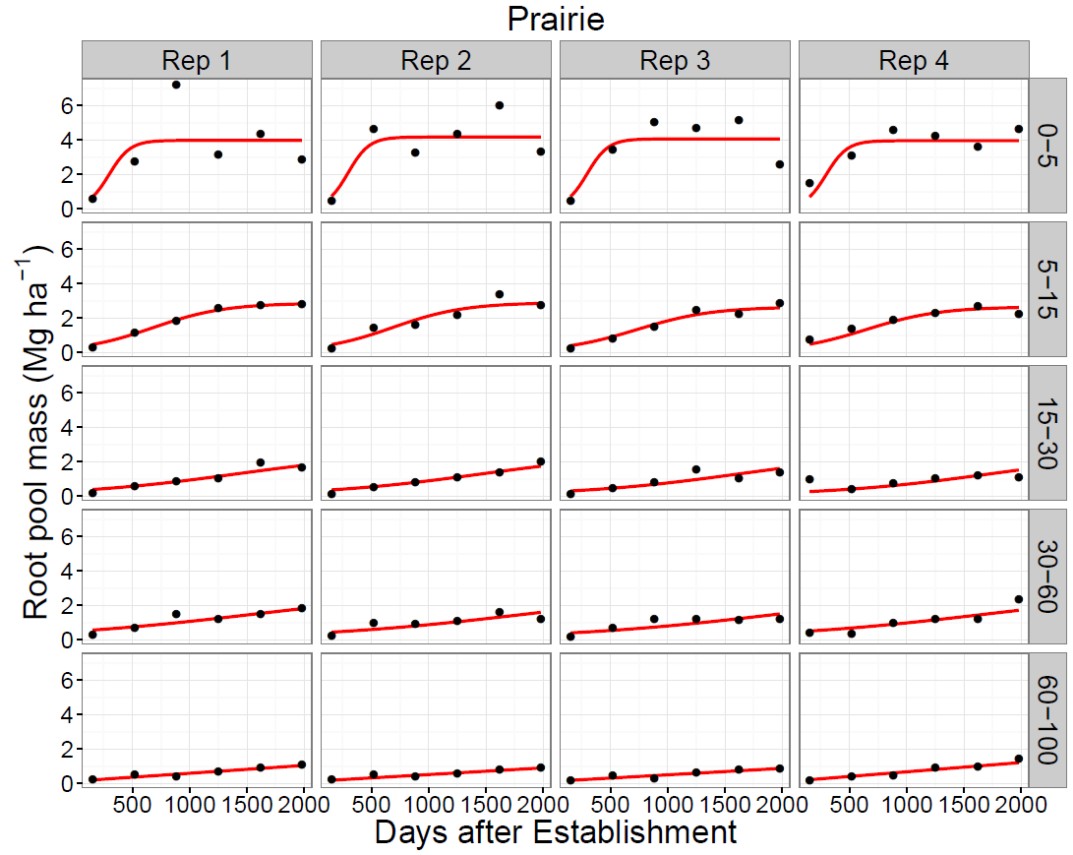

Fig A1. Logistic curves fit to root pool mass accumulation at each replication and depth increment in the prairie treatment.





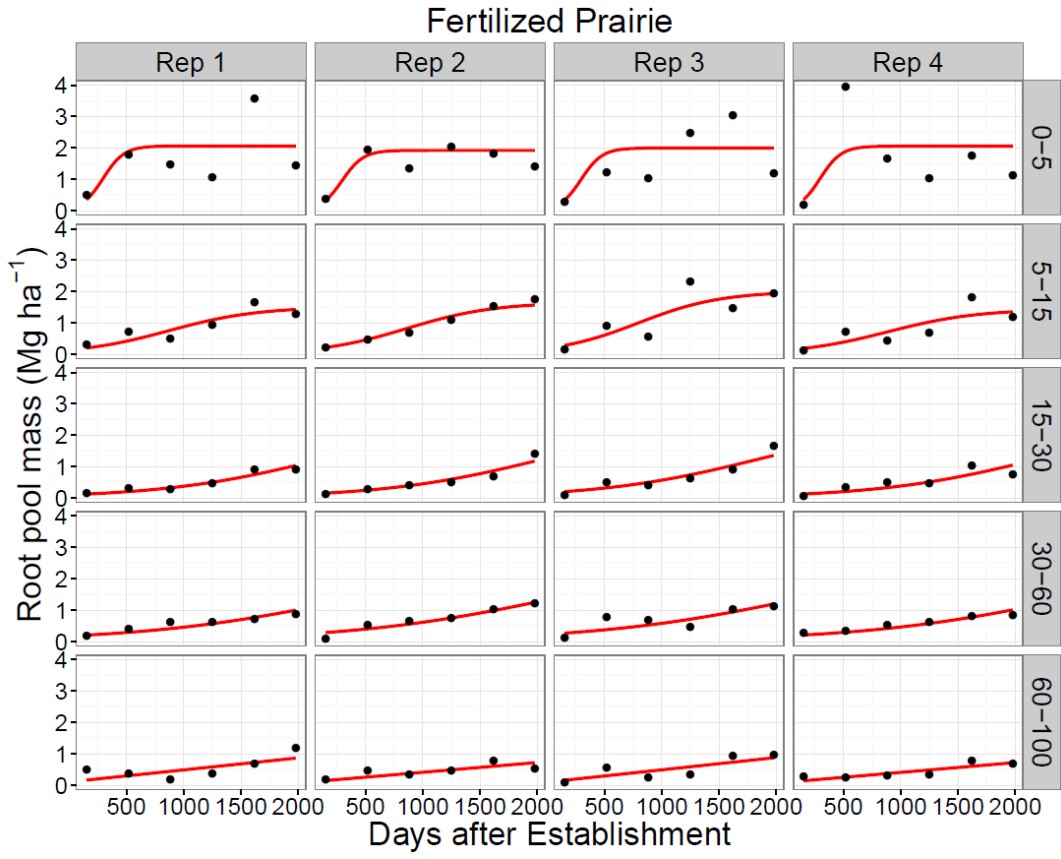

Fig A2. Logistic curves fit to root pool mass accumulation at each replication and depth increment in the fertilized prairie treatment.





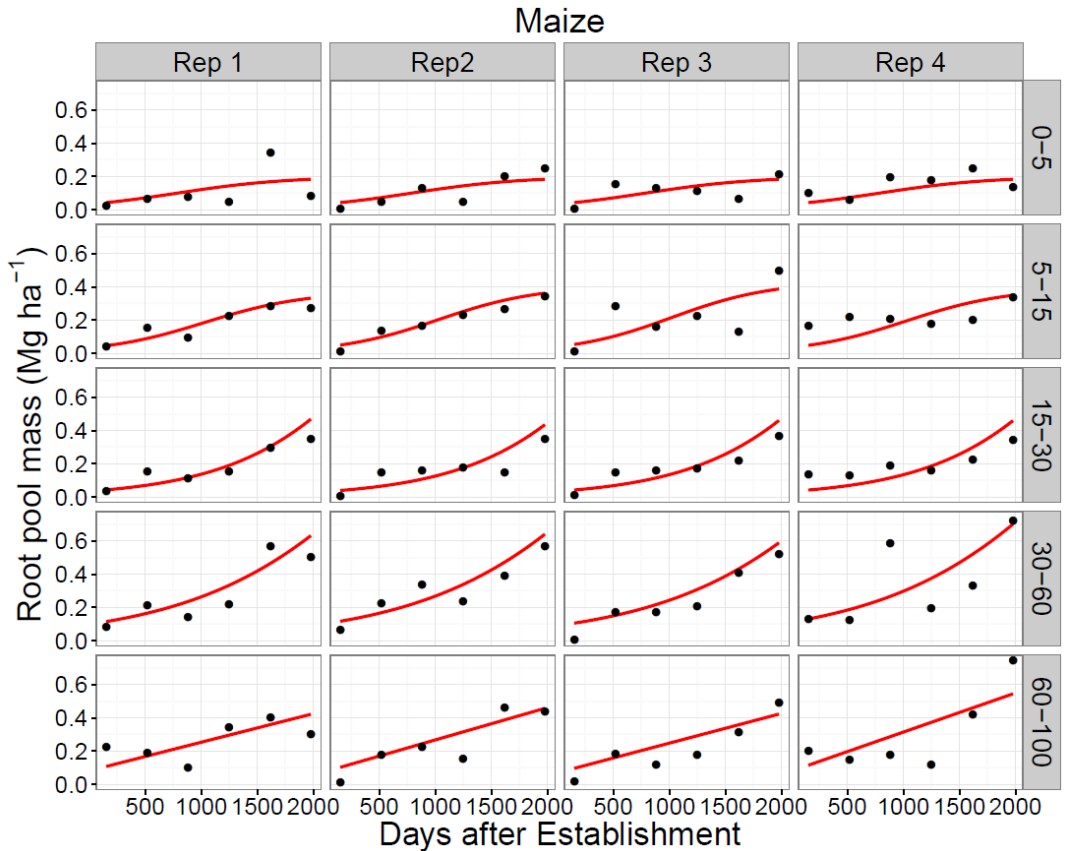

Fig A3. Logistic curves fit to root pool mass accumulation at each replication and depth increment in the maize treatment.

**Competing interests**

The authors have no competing interests.

**Acknowledgements**

This project was supported by Agriculture and Food Research Initiative Competitive Grant numbers 2012-67011-1966 and 2016-67012-
25170 from the USDA National Institute of Food and Agriculture and Iowa State University's College of Agriculture and Life Sciences,
Department of Agronomy, Graduate Program in Sustainable Agriculture, and Wallace Chair for Sustainable Agriculture. We thank Dave
Sundberg, Bruce Hall, Kevin Day, Jake Anderson, Brent Beelner, Shane Bugeja, Robin Gómez-Gómez, Céline Guignard, Sarah Hirsh,
Brady North, Nick Siepker, and Madeline Tomka for technical assistance in the field and laboratory. We are also grateful to Lendie
Follett and the R community for help with statistics and coding as well as several anonymous reviewers for useful comments on earlier
drafts of the manuscript.



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
