# Peer review of "A deeper look at the relationship between root carbon pools and the vertical distribution of the soil carbon pool"

_SOIL, 2017_

## Referee Comment (RC1) · Anonymous Referee #1 · 11 Apr 2017

In this study, Dietzel et al. used an agronomic trial to study the linkage between root C input and the vertical distribution of Soil Organic Matter (SOM). Using a soil under corn cultivation for more than a century, they measured the SOM profile of this soil as well as the root material input and quality (C:N ratio) along the soil depth for both prairie and maize vegetation. They found that maize allocates a higher proportion of root input in deep soil layers and that it as a lower C:N ratio compared to prairie plants, which is quite classical. Further, they found that root C:N ratio increases with depth for all the treatments. This result is interesting and quite new from what I know. This suggests that deeper roots are dominated by transport root with highly sclerified tissues and poor absorptive proteins content compared to surface roots. Finally, they conclude that

in moving from prairie to maize, a large, structural-tissue dominated root C pool with slow turnover, concentrated at shallow depths was replaced by a small, non-structural-tissue dominated root C pool with fast turnover evenly distributed in the soil profile, suggesting that maize may allocate more root C input to the soil than prairies at deeper depths. This constitutes the strong portion of this manuscript.

Based on the conceptual framework and the empirical results of the study of Cotrufo et al. (2015) about the formation of SOM, they also argue that their pattern of increasing root C:N ratio with depth could explain why an disproportionally large stock of SOM relative to root C inputs is found in deep soil. First, I found it quite tricky to conclude about the driver of such a global scale pattern from data of a case study like this. Beyond that, I am not convinced by this interpretation and I found their argumentation about this statement quite weak for several reasons related to logical contradiction and some misunderstanding about the work of Cotrufo as discussed into more detail below. If I consider these two paths of SOM formation and your results together, I would consider that shallow root of high litter quality would supply high input of DOC that can be efficiently processed by soil microorganisms (high Carbon Use Efficiency [CUE]) and supply larger quantity of microbial by-products that can be then stabilized in soil microaggregates by mineral-binding, thus leading to higher C sequestration. In contrast, the deep root of poor quality (higher proportion of POM) will be least efficiently processed, thus leading to higher C lost by mineralization relative to SOM formation and ultimately lower C sequestration. This is thus not consistent with the pattern of the disproportionally large stock of SOM relative to root C inputs in deep soil. Further, Cotrufo et al. (2015) studied SOM formation over short-term scale whereas deep soil C is often hundreds to thousands year-old and highly microbialy processed. Fontaine et al. (2007) found that deep soil C mineralization is strongly limited by energetic constraints. This slow turnover together with the DOC input from surface to deep soil layer documented by Rumpel and Kögel-Knabner (2010) could more likely explain the disproportionally large stock of SOM relative root C input is found in deep soil. I also pointed several methodological issues detailed below. Finally, I fell that you did not

so much discussed how the root system of your different plant communities (maize vs. prairie) could explain the vertical C profile of your studied soil though this constitutes the strong part of your study to the linkage between root C input and the vertical distribution of SOM.

Taken together, I think this manuscript will need important revisions to be acceptable for publication, especially by avoiding tricky extrapolation and misinterpretation and by refocused on the conclusion you can reasonably draw from your results. Clarify your scientific questions/hypotheses could also help to achieve this end.

  Detailed discussion of the manuscript P.1-L. 15. 'in moving from prairie to maize' If I well understood your design, you studied soil root allocation on restored prairies that have maize cropping historical of >100 years. Therefore, would it not be more correct to talk about moving from maize to prairie. P.1-L. 15. 'contribute' to what? To soil C stock? Please clarify. Alternatively, we could also talk about 'C allocation'. P.1-L. 21. Please clarify what you mean by 'aboveground process'. Is really soil disturbance (tillage?) an aboveground process? P.1-L. 26-27. Is this definition really necessary here? I think it will be better placed in the Material and Methods section. P.2-L.5. Please insert the Weaver citation P.2-L.17. Why did you used 'Carbon:N' though you used 'C:N' just before. P.2-L.19-21. I do not clearly see how your experimental design give you a 'unique perspective on characteristics of root inputs' Please clarify. It would also be useful to indicate here the number of year since prairie restoration at the end of the study (5 years not?). P.2-L.26-27. I did not understand the point of your second scientific question before to read the last extion of your discussion. Please be not explicit and precise on your purpose. P.3-L.1-2. What about soil N concentration? This could be importabnt P.3-L.1-2. How many replicates (blocks)? 4? This information is crucial! P.5-L.4-8. You used linear mixed models. Please state what factors are formulated as fixed or random effects in your models. P.5-L.15-23. Logistic model is used fit binary response variable. Therefore, I do not see the rationale to use Logistic model to fit root mass, which is a continuous variable. . . P.5-L.19. It is not so clear to me what you mean by "root

mass accumulation". It is the difference in root mass between two sampling dates? Or is it cumulative root growth? But you did not measure it between all sampling dates, right? Please clarify. By the way, it not so clear what was the initial root mass stock and distribution prior to experimental set-up. P.5-L.29-30. We calculated root turnover constant as k = root loss / root stock. This computation is quite uncommon. Hence, Gill and Jackson (2000) calculated root turnover constant as k = root gain / root stock. In addition to be more standard method, I also found it clearer as root gains are directly obtained with the ingrowth core method while your root loss computation use root mass accumulation, which was not very well defined. P.7-L.9. Throughout the manuscript, we heavily use the 'pool'. Though I found this term appropriate for distingue different component of the global soil C stock, I found the term 'stock' more suitable when talking about quantitative estimate. P.6-L.6. There is no reference to Table 1 in the text. P.11-12. There is no reference to most of your tables and figures in this portion of your result section... You really need to clearly use reference to it for justify what you state in the text. In its current state, I do feel really difficult to follow your text. P.10-Table 3. Is this really useful ? Figure 4 already provide this information. This table should be place in appendix. By the way, I found that there is quite too much table and figure in the article. P.11-L.15. What you mean by input? Is it your root mass accumulation? Please clarify. P.12-L.6. What about soil N concentration and soil C:N ratio across soil depth and treatment? Isn't this information important is understand the root C:N profiles? P.13-L7-8. 'a physical-transfer pathway whereby plant tissue is processed by soil microbes to its fullest extent, and then remains in the soil functionally inert'. Really? Cotrufo et al. (2015) actually talk about physical transfer of Particulate Organic Matter (POM) from litter to soil. POM is not functionally inert! P.13-L.11-19. 'root decomposition in our study would have resulted in a gradient of microbially-derived to physically-derived organic matter from the top of the soil profile downward' Then this is not consistent with evidence that the contribution of microbial- and not root-derived C increases with depth (Rumpel and Kogel-Knabner, 2011) in contrast with what you stated L.15-16. I assume that DOC derived from soil surface can be mobile and move down the profile but a large portion can be stabilized in the surface and at least the SOM derived from deep root with high C:N ratio should be less microbial-derived given what you state. This point should be clarify. Moreover, the notion physically-derived SOM does not make sense, see my previous comment. P.13-L.12-14. 'Soil organic matter at the soil surface would be vulnerable to transport to greater depth as dissolved organic C whereas physically-transferred soil organic matter at depth would be relatively immobile'. If you read carefully Cotrufo (2015), she stated that DOC derived from litter is preceded by soil microorganisms and the microbial by-products are then stabilized in soil microagregates by mineral-binding. This mineral-stabilized SOM is thus actually less mobile than POM, in contrast with what you stated. P.13-L.16-19. Exsudates are highly labile compounds that are very quickly preceded by soil microorganisms. Once metabolized, they are much less mobile. Therefore, they probably represent a minor fraction DOC moving down the profile and that could form deep SOC. P.13-L.27-29. 'By the sixth year of reconstructed prairie establishment, root C pool equilibrium was reached and prairies began making substantial annual contributions to the soil organic matter pool above 30 cm, although the fraction of organic matter that remained in the soil is unknown' You have information on root litter decomposition and soil organic matter turnover, so cannot state anything about SOM formation or stock. All you can saw this you likely have higher root litter input that could eventually increase SOM stock. P.13-L.35-37. Probably, but this is quite speculative... P.14-L.10. 'contributed more C' This is unclear.   References Cotrufo, M. F., Soong, J. L., Horton, A. J., Campbell, E. E., Haddix, M. L., Wall, D. H. & Parton, W. J. (2015) Formation of soil organic matter via biochemical and physical pathways of litter mass loss. Nature Geosci, 8, 776-779. Fontaine, S., Barot, S., Barre, P., Bdioui, N., Mary, B. & Rumpel, C. (2007) Stability of organic carbon in deep soil layers controlled by fresh carbon supply. Nature, 450, 277-U10. Gill, R. A. & Jackson, R. B. (2000) Global patterns of root turnover for terrestrial ecosystems. New Phytologist, 147, 13-31. Rumpel, C. & Kögel-Knabner, I. (2010) Deep soil organic matter—a key but poorly understood component of terrestrial C cycle. Plant and Soil, 338, 143-158.

---

## Short Comment (SC1) · 11 Apr 2017

*A note upfront from the submitting person: This review was prepared by Nadja Huber and Mirjam Mächler, both master students in geography at the University of Zurich. The review was part of an exercise during a second semester master level seminar on "the biogeochemistry of plant-soil systems in a changing world", which I organize. We would like to highlight that the depth of scientific knowledge and technical understanding of these reviewers represents that of master students. We enjoyed discussing the manuscript in the seminar, and hope that our comments will be helpful for the authors.*

Dietzel et al. start with the fundamental statement, that soil organic carbon and root mass are disproportionately distributed in soils, supposing that root mass has a direct

influence on soil carbon pool. As a matter of fact, in a depth below 20cm half of all soil organic C in soils can be found where just a third of all root mass is. There is no clear answer to the question, why there is such a large difference between the two C pools. Dietzel et al. mention that temperature, moisture, O2, soil texture and soil C values are part of the explanation of this discrepancy. Still, the C:N ratio as part of it has always been neglected in previous studies.

The paper therefore specifically concentrates on a more detailed look at the properties of C pools. For this purpose the authors examined soil C and root C pools in three different cropping systems. Continuous maize, multispecies prairie and N-fertilized multispecies prairie. Research questions asked were the following: "1) How does the quantity, distribution, and C:N ratio of the root C pool differ with depth and between these native perennial and non-native annual ecosystems and 2) what do these differences in input tell us about the historical belowground ecosystem under which these soils developed and the systems and will these soils continue to change?" To answer these questions the authors conducted a field experiment over six years in Boone County, IA, USA. With this field experiment the authors were able to show that an increase in root C:N ratio with depth is a potentially important factor determining the distribution of C in the soil profile. The authors consider the root pool C:N ratios to be sufficiently important that they result in a greater maize C contributions to soil organic matter than prairie C below 20 cm.

Objective 1 (root quality and quantity with depth) was discussed in detail in the manuscript. During the discussion in the classroom, however, it became clear that objective 2 and the related discussion confused all of us. We did not understand i) why the "historical belowground ecosystem" is important and ii) how objective 2 relates to the presented results. The question if these soils will continue to change is rhetoric (soil always continue to develop) and very unspecific. The corresponding discussion section (4.3) is very short and speculative. Are root "turnover" (with a lifetime of a few years) and soil organic carbon turnover (decades to centuries) somehow related?

Probably they are not. Coincidence is not causation. We wondered if objective 2 and section 4.3 are needed at all. If you think they are, please elaborate this part of the manuscript.

Detailed comments: We did not fully understand the link between C:N and root depth. Do you mean that C:N ratios increase with depth depending on species or on individual plants?

P. 3, line 26: what does "sampling by replicate block from 31 October-25 November 2008" mean? Did you sample repeatedly? Explanation of "replicate block"-approach needed.

P. 4, line 11/12: is there a difference between root measurements and root data?

P. 4, line 24/25: why different storage? Further explanation desired.

P. 5, line 28: why the period between April 1st and November 30th to calculate the average root mass accumulation? Are these official dates? Further explanation needed.

Table p. 10: unclear -> explanation of upper/lower case letters and meaning of those letters is missing; it could be part of the description of the table

Explanation pro glimmix on page 5/14 but not in table description. P. 13, line 3: increase in root pool C:N ratio has not been reported previously in the literature:

We would appreciate some information about previous research which focused on a related topic

P. 14, line 3/4: the pattern of distribution of what? Do you mean the vertical distribution of roots? What is place in this context?

P. 14, line 30-32: For us, this sentence is very long and difficult to understand. No significant changes in soil C (changes in quantity or stocks?) at any depth but differences in quantity? "implementation of annual cropping systems": Do you refer to line 19? Experimental location was a site of cultivation under annual crops for over 100 years.

**SOILD**

Remarks concerning formal structures (typos, figures etc.): P. 3, line 2: typo: 11 mg kg-1

Figure p. 6: why not making a title with total C, root C maize, etc. instead of letters a,b,c -> would be more clear and consistent with the following figures

Figures p.8 + 9: legend can be improved -> no units -> unclear & colors are not suitable for black/white printing

P. 13, line 36: typo: this

---

## Referee Comment (RC2) · Anonymous Referee #2 · 17 Apr 2017

Dietzel et al. report on a root study conducted at a field experiment where continuous corn is compared to reconstructed fertilized and unfertilized prairie stands. They have measured: 1) root profiles to depth of 1 m at the end of the growth season for six consecutive years, 2) root production (by regrowth cores) for 2 growing seasons to 30 cm depth, 3) root and soil C and N concentrations to 1 m depth. Extracting root for multiple growing seasons, multiple soils layers and multiple replicated treatments is by no mean easy, and the soil science community can certainly benefit from such precious data. The authors report interesting findings: 1) the C/N ratio of root material increases with depth, which has potential implications for soil C storage, 2) the maize root profile is more uniform with depth than that of prairie species (confirmative), 3) fertilization of

the reconstructed prairie greatly decreases root biomass.

However, I have some significant concerns with the study:

1) The continuous accumulation of maize roots throughout the 6-year period is quite troubling (Fig A3). The authors provide one reference (Dupont 2014) stating that intact prairie root (not maize) can be found in soil several years after cultivation. However, they ignore the substantial literature on maize roots that clearly indicates that maize roots decompose rapidly in soils, starting with the classical study of Mengel and Barber in 1974 (Agron. J. 66: 341-344) and several studies that have followed. Actually, Mengel and Barber (1974) state that root length and fresh weight decrease rapidly after maize has reached the reproductive stage. Here, Dietzel et al. themselves state in the abstract about maize roots that they are "non-structural-tissue dominated root C pool with fast turnover". They also indicate that the site was apparently under maize-soybean rotation prior to starting the experiment, so why was there no accumulated maize root biomass at the start of the experiment (if the root accumulation theory is correct)? Unfortunately, in the present study the roots were not sampled at the same time each year (from early October to early November), and the accumulation of maize roots the last two years also corresponds to the 2 earliest sampling. A possible explanation is that the roots actually decomposed quickly in the field and that by sampling a month earlier by the end of the six-year period a greater number of non-decomposed roots were retrieved. Effects of inter-annual climatic variability on root growth is another potentially contributing effect. There are three implications from this: 1) apparent maize-root accumulation in the field over 6 years is probably an artefact, 2) the pool and rate modelling of Fig 3 and 4 is not justified (it did not bring much to the paper anyway), 3) the paper should have included a much more throughout review of the literature about maize-root dynamics in field soils.

2) The maize-root C profile is presented 3 times in the paper: 1) Fig 1 b, 2) Fig 2, and 3) Fig. A3. The 3 figures are in the same units (Mg ha-1), but I could not reconcile the data between them. The 2013 data of fig. 1 b (with largest root accumulation in

the top soil) do not seem to correspond to the 2013 data of figure A3, which seems to show highest maize-root biomass in the deeper soil. In addition, the maize root profile appears more even in Fig.2 than in Fig. 1 b, while it should be exactly the opposite (e.g. the 5-10 cm should have about half of the 0-5 cm in fig 2, if extrapolated from fig 1). Or are the two figures exactly the same? But why figure 2 then? The 3 figures should have been reconciled and presented as one main figure in logical units, and then the results compared to the literature.

3) Data from the root regrowth cores are not clearly presented, but used for a direct extrapolation of a root turnover rate in the top 30 cm. Summary data (without statistics) are presented in g m-2 in Table 4, making it difficult to compare to the root pool data presented in Mg ha-1. The maize root productivity appears low and I am missing a coherent evaluation of the C input in the context of published studies.

4) The implications for C storage presented in this paper are largely hypothetical and somewhat contradictory. The present paper contains no significant result to link root biomass profiles to soil C profiles. While it is OK to briefly elaborate about possible implication of a higher root C/N ratio with depth, this should not be the main part of the discussion, which should instead focus on actual significant results. In addition, I could not reconcile the two ideas presented here about the effect of root C/N ratio on C storage in soils. On the hand, the authors argue that a lower C/N ratio for maize root favours C storage in soil as compared to prairie roots (p14, line 11-12). On the other hand, they also argue that an increasing C/N ratio of roots with depth in the soil profile also favours C storage (e.g. p1, line 10-12). The potential attempt to reconcile these two contradictory effects of root C/N ratio on soil C was unconvincing (p 14).

In conclusion, Dietzel et al. have collected an impressive data set on maize and prairie roots following maize-soybean rotation. The dataset appears to suffer from some arte-facts, but root studies are difficult and shortcomings could have been acknowledged. The data themselves are neither clearly presented nor sufficiently discussed in light of the literature. A main finding is largely ignored, i.e. the dramatic reduction of root

biomass by fertilization in prairie systems. By contrast, the authors focus on an uncertain modelling and non-verifiable considerations about the effect of root C/N ratio on soil C. A focus on significant results and discussion of these results in light of the literature would have better served this study.

---

## Author Comment (AC1) · 19 May 2017

In this study, Dietzel et al. used an agronomic trial to study the linkage between root C input and the vertical distribution of Soil Organic Matter (SOM). Using a soil under corn cultivation for more than a century, they measured the SOM profile of this soil as well as the root material input and quality (C:N ratio) along the soil depth for both prairie and maize vegetation. They found that maize allocates a higher proportion of root input in deep soil layers and that it as a lower C:N ratio compared to prairie plants, which is quite classical. Further, they found that root C:N ratio increases with depth for all the

treatments. This result is interesting and quite new from what I know. This suggests that deeper roots are dominated by transport root with highly sclerified tissues and poor absorptive proteins content compared to surface roots. Finally, they conclude that in moving from prairie to maize, a large, structural-tissue dominated root C pool with slow turnover, concentrated at shallow depths was replaced by a small, non-structural tissue dominated root C pool with fast turnover evenly distributed in the soil profile, suggesting that maize may allocate more root C input to the soil than prairies at deeper depths. This constitutes the strong portion of this manuscript.

Thank you, we too were excited to quantify differences in prairie and maize root allocation and find root C:N ratios increase with depth.

Based on the conceptual framework and the empirical results of the study of Cotrufo et al. (2015) about the formation of SOM, they also argue that their pattern of increasing root C:N ratio with depth could explain why an disproportionally large stock of SOM relative to root C inputs is found in deep soil. First, I found it quite tricky to conclude about the driver of such a global scale pattern from data of a case study like this.

Yes, since this is only one study, we avoid making any strong conclusions, but do suggest that root C:N ratio plays a role in development of the soil profile. Finding increases in root C:N ratios with depth is significant on its own, but we feel it is very important to put this finding in the context of larger scientific questions.

Beyond that, I am not convinced by this interpretation and I found their argumentation about this statement quite weak for several reasons related to logical contradiction and some misunderstanding about the work of Cotrufo as discussed into more detail below.

We feel we can do a better job communicating our proposed mechanism and we address these details below. We found your comments to be extremely helpful towards improving this manuscript and enjoyed getting a new perspective on many of the aspects we have wrestled with during writing. It seems we all have the same understanding of Cortrufo et al.'s conceptual models, but are used to thinking about these

models under different environmental circumstances. One major assumption on which we do not completely agree is the likeliness of microbial by-products to be transported deeper in the soil profile. We hope the discussion below and additional references help to clarify the manuscript.

If I consider these two paths of SOM formation and your results together, I would consider that shallow root of high litter quality would supply high input of DOC that can be efficiently processed by soil microorganisms (high Carbon Use Efficiency [CUE]) and supply larger quantity of microbial by-products that can be then stabilized in soil microaggregates by mineral-binding, thus leading to higher C sequestration. In contrast, the deep root of poor quality (higher proportion of POM) will be least efficiently processed, thus leading to higher C lost by mineralization relative to SOM formation and ultimately lower C sequestration. This is thus not consistent with the pattern of the disproportionally large stock of SOM relative to root C inputs in deep soil.

Right! We find this inconsistency in patterns very interesting and a major motivation for the manuscript. Although the proposed relationship we describe between root C:N ratio and soil C profile development is not immediately intuitive, the combination of MEMS and dissolved organic carbon (C) transport leads to a very possible mechanism behind a disproportionally large stock of SOM relative to root C inputs.

Further,Cotrufo et al. (2015) studied SOM formation over short-term scale whereas deep soil C is often hundreds to thousands year-old and highly microbialy processed.

Yes, Cortrufo et al. have focused on short-term scales and their conceptual model is still hypothetical, however, what happens in the short-term is directly connected to what happens in the long term. The fact that deep soil C is highly microbially processed does not indicate where where the C originated.

Fontaine et al. (2007) found that deep soil C mineralization is strongly limited by energetic constraints. This slow turnover together with the DOC input from surface to deep soil layer documented by Rumpel and Kögel-Knabner (2010) could more likely explain

the disproportionally large stock of SOM relative root C input is found in deep soil.

Both of these factors, and many more mentioned in the manuscript, contribute to the disproportionally large stock of SOM relative to root C input found in deep soil, but do not fully explain it and do not incorporate the role of root C:N ratio. The transport of DOC is especially important in our proposed mechanism and we spend some time on how roots at shallow depths vs. deeper depths contribute to this DOC.

I also pointed several methodological issues detailed below. Finally, I fell that you did not so much discussed how the root system of your different plant communities (maize vs. prairie) could explain the vertical C profile of your studied soil though this constitutes the strong part of your study to the linkage between root C input and the vertical distribution of SOM.

Thank you for the methodological questions below. We would have liked to spend more discussion on the root systems of maize vs. prairie, but felt that without measurements of the original soil C profile, discussion specific to change created by annual cropping systems would be challenged. However, we can strengthen this component in response to your comment.

Taken together, I think this manuscript will need important revisions to be acceptable for publication, especially by avoiding tricky extrapolation and misinterpretation and by refocused on the conclusion you can reasonably draw from your results. Clarify your scientific questions/hypotheses could also help to achieve this end.

Detailed discussion of the manuscript

P.1-L. 15. 'in moving from prairie to maize' If ?? I well understood your design, you studied soil root allocation on restored prairies that have maize cropping historical of >100 years. Therefore, would it not be more correct to talk about moving from maize to prairie.

We are referencing the historical shift from prairie to maize. I will change the wording

for clarification.

P.1-L. 15. 'contribute' to what? To soil C stock? Please clarify. Alternatively, we could also talk about 'C allocation'.

Yes, soil C stock. We will clarify

P.1-L. 21. Please clarify what you mean by 'aboveground process'. Is really soil disturbance (tillage?) an aboveground process?

Will change 'aboveground processes' to 'soil management'.

P.1-L.26-27. Is this definition really necessary here? I think it will be better placed in the Material and Methods section.

We included it here as we go on to use the definition in the introduction.

P.2-L.5. Please insert the Weaver citation

Thanks for catching this, we will insert the citation.

P.2-L.17. Why did you used 'Carbon:N' though you used 'C:N' just before.

We did not want to start a sentence with an abbreviation.

P.2-L.19-21. I do not clearly see how your experimental design give you a 'unique perspective on characteristics of root inputs' Please clarify.

We expect many of the characteristics reported here to be less detectable in well-established prairies systems, but you are right that prairie reconstruction in not entirely unique. We will replace "unique perspective" to "new perspective".

It would also be useful to indicate here the number of year since prairie restoration at the end of the study (5 years not?).

Yes, this will be added here.

P.2-L.26-27. I did not understand the point of your second scientific question before

to read the last extion of your discussion. Please be not explicit and precise on your purpose.

Thanks for this comment. You illustrate that we need to change some aspects of the introduction to make it understandable for an international audience. For example, in this instance we will simply say "perennial prairie" instead of "historical" and "annual cropping systems" instead of "current systems".

P.3-L.1-2. What about soil N concentration? This could be importabnt

We do have total soi N data and will include it here.

P.3-L.1-2. How many replicates (blocks)? 4? This information is crucial!

I'm sorry we did not include this, it will be added.

P.5-L.4-8. You used linear mixed models. Please state what factors are formulated as fixed or random effects in your models.

This information will be added.

P.5-L.15-23. Logistic model is used fit binary response variable. Therefore, I do not see the rationale to use Logistic model to fit root mass, which is a continuous variable. . .

We will add more details here. Logistic regression is used to fit binary response variables. We used a logistic function to fit the data and then statistically compared the parameters of each fit of the function as described in the book "Mixed Effect Models in S and S-plus", Pinheiro and Bates, 2000. We will add this citation.

P.5-L.19. It is not so clear to me what you mean by "root mass accumulation". It is the difference in root mass between two sampling dates? Or is it cumulative root growth? But you did not measure it between all sampling dates, right? Please clarify. By the way, it not so clear what was the initial root mass stock and distribution prior to experimental set-up.

"Root mass accumulation" refers to root mass gained. We did not measure it between all sampling dates, rather we used the model we fit to predict values during times we did not make measurements.

P.5-L.29-30. We calculated root turnover constant as k = root loss / root stock. This computation is quite uncommon. Hence, Gill and Jackson (2000) calculated root turnover constant as k = root gain / root stock. In addition to be more standard method, I also found it clearer as root gains are directly obtained with the ingrowth core method while your root loss computation use root mass accumulation, which was not very well defined.

We can and will easily replace "root loss" with "root gain" in our equation.

P.7-L.9. Throughout the manuscript, we heavily use the 'pool'. Though I found this term appropriate for distingue different component of the global soil C stock, I found the term 'stock' more suitable when talking about quantitative estimate.

We will change the text so that 'pool' is used only when discussing specific components of the global soil C stock.

P.6-L.6. There is no reference to Table 1 in the text.

Our apologies, we will correct this.

P.11-12. There is no reference to most of your tables and figures in this portion of your result section. . . You really need to clearly use reference to it for justify what you state in the text. In its current state, I do feel really difficult to follow your text.

We will add more references to make the text easier to follow.

P.10-Table 3. Is this really useful ? Figure 4 already provide this information. This table should be place in appendix. By the way, I found that there is quite too much table and figure in the article.

We will be happy to move Table 3 to the appendix and consider removing or combining

some of the figures.

P.11-L.15. What you mean by input? Is it your root mass accumulation? Please clarify.

Root mass accumulation is root mass input - root mass loss. Input is how much root mass went into the soil. We will clarify this in the text.

P.12-L.6. What about soil N concentration and soil C:N ratio across soil depth and treatment? Isn't this information important is understand the root C:N profiles?

Yes, this information is important and we can include total (organic + inorganic) N values for this soil. While root C:N ratio increases with depth, soil C:N ratio decreases with depth, and it may be useful to discuss this relationship.

P.13-L7-8. 'a physical-transfer pathway whereby plant tissue is processed by soil microbes to its fullest extent, and then remains in the soil functionally inert'. Really? Cotrufo et al. (2015) actually talk about physical transfer of Particulate Organic Matter (POM) from litter to soil. POM is not functionally inert!

We will change this to better reflect Cotrufo's original language and refer instead to the "inherent chemical recalcitrance" of organic matter resulting from the physical-transfer pathway.

P.13-L.11-19. 'root decomposition in our study would have resulted in a gradient of microbially-derived to physically-derived organic matter from the top of the soil profile downward' Then this is not consistent with evidence that the contribution of microbial- and not root-derived C increases with depth (Rumpel and Kogel-Knabner, 2011) in contrast with what you stated L.15-16.

The sentence following this one in the manuscript is very important. The microbially-derived organic matter would be mobile and transported to deeper depths, contributing to the relatively immobile pool of physically-derived organic matter. This is very consistent with Rumpel and Kogel-Knaber (2011), as we eventually conclude.
I assume that DOC derived from soil surface can be mobile and move down the profile but a large portion can be stabilized in the surface and at least the SOM derived from deep root with high C:N ratio should be less microbial-derived given what you state. This point should be clarify. Moreover, the notion physically-derived SOM does not make sense, see my previous comment.

Yes, DOC can be stabilized in the soil surface, but the proportion of C stabilized depends on soil type and level of C saturation. We will add a reference to Castellano (2015) to support this idea. These prairie-formed soils do not have C concentrations as high as historical levels, but total C concentrations are still at 2.8 percent, indicating a reduced capacity for additional C stabilization. In this environment, microbial by-products are likely to be part of the soil solution and easily transported to deeper depths with greater capacity for C stabilization. We will make this clearer in the manuscript.

P.13-L.12-14. 'Soil organic matter at the soil surface would be vulnerable to transport to greater depth as dissolved organic C whereas physically-transferred soil organic matter at depth would be relatively immobile'. If you read carefully Cotrufo (2015), she stated that DOC derived from litter is preceded by soil microorganisms and the microbial by-products are then stabilized in soil microaggregates by mineral-binding. This mineral-stabilized SOM is thus actually less mobile than POM, in contrast with what you stated.

Microbial by-products are very mobile until they are stabilized. When and where they are stabilized depends on the soil conditions. In the mechanism we propose, microbial by-products reach the deeper profile and are stabilized there. This is consistent with findings that proportion of microbial-derived SOM increases with depth.

P.13-L.16-19. Exsudates are highly labile compounds that are very quickly preceded by soil microorganisms. Once metabolized, they are much less mobile. Therefore, they probably represent a minor fraction DOC moving down the profile and that could form deep SOC.

[Figure]

Thank you, we should provide clarification that exudates quickly move into microbial pools. However, the fate of the C after that again depends on the environmental capacity for microbial by-products to be stabilized.

P.13-L.27-29 'By the sixth year of reconstructed prairie establishment, root C pool equilibrium was reached and prairies began making substantial annual contributions to the soil organic matter pool above 30 cm, although the fraction of organic matter that remained in the soil is unknown' You have information on root litter decomposition and soil organic matter turnover, so cannot state anything about SOM formation or stock. All you can saw this you likely have higher root litter input that could eventually increase SOM stock.

You are absolutely right. This will be fixed by changing 'contributions' to 'inputs'.

P.13-L.35-37. Probably, but this is quite speculative. . .

It is indeed speculative, but the most reasonable answer given the evidence available.

P.14-L.10. 'contributed more C' This is unclear.

Will change to 'had greater C inputs'.

References

Cotrufo, M. F., Soong, J. L., Horton, A. J., Campbell, E. E., Haddix, M. L., Wall, D. H. & Parton, W. J. (2015) Formation of soil organic matter via biochemical and physical pathways of litter mass loss. Nature Geosci, 8, 776-779.

Fontaine, S., Barot, S., Barre, P., Bdioui, N., Mary, B. & Rumpel, C. (2007) Stability of organic carbon in deep soil layers controlled by fresh carbon supply. Nature, 450, 277-U10.

Gill, R. A. & Jackson, R. B. (2000) Global patterns of root turnover for terrestrial ecosystems. New Phytologist, 147, 13-31.

[Figure]

Rumpel, C. & Kögel-Knabner, I. (2010) Deep soil organic matter a key but poorly understood component of terrestrial C cycle. Plant and Soil, 338, 143-158.

Castellano, M. J., Mueller, K. E., Olk, D. C., Sawyer, J. E. and Six, J.: Integrating plant litter quality, soil organic matter stabilization, and the carbon saturation concept, Global Change Biology, 21(9), 3200-3209, doi:10.1111/gcb.12982, 2015.

---

## Author Comment (AC2) · 19 May 2017

M. W. I. Schmidt michael.schmidt@geo.uzh.ch

*A note upfront from the submitting person: This review was prepared by Nadja Huber and Mirjam Mächler, both master students in geography at the University of Zurich. The review was part of an exercise during a second semester master level seminar on "the biogeochemistry of plant-soil systems in a changing world", which I organize. We would like to highlight that the depth of scientific knowledge and technical understanding of these reviewers represents that of master students. We enjoyed discussing the manuscript in the seminar, and hope that our comments will be helpful for the authors.*

[Figure]

That is so great that you discussed our paper as a group! What a great exercise and these comments are just the sort of thing we hoped would happen when submitting a paper to Soil. Thanks a lot for your comments. Hearing from many people really makes obvious which portions of the paper are difficult to understand and where the most improvement is needed.

Dietzel et al. start with the fundamental statement, that soil organic carbon and root mass are disproportionately distributed in soils, supposing that root mass has a direct influence on soil carbon pool. As a matter of fact, in a depth below 20cm half of all soil organic C in soils can be found where just a third of all root mass is. There is no clear answer to the question, why there is such a large difference between the two C pools. Dietzel et al. mention that temperature, moisture, O2, soil texture and soil C values are part of the explanation of this discrepancy. Still, the C:N ratio as part of it has always been neglected in previous studies.

Indeed, decreases in root C:N ratio with depth has been an unknown factor.

The paper therefore specifically concentrates on a more detailed look at the properties of C pools. For this purpose the authors examined soil C and root C pools in three different cropping systems. Continuous maize, multispecies prairie and N-fertilized multispecies prairie. Research questions asked were the following: "1) How does the quantity, distribution, and C:N ratio of the root C pool differ with depth and between these native perennial and non-native annual ecosystems and 2) what do these differences in input tell us about the historical belowground ecosystem under which these soils developed and the systems and will these soils continue to change?" To answer these questions the authors conducted a field experiment over six years in Boone County, IA, USA. With this field experiment the authors were able to show that an increase in root C:N ratio with depth is a potentially important factor determining the distribution of C in the soil profile. The authors consider the root pool C:N ratios to be sufficiently important that they result in a greater maize C contributions to soil organic matter than prairie C below 20 cm.

Objective 1 (root quality and quantity with depth) was discussed in detail in the manuscript.

During the discussion in the classroom, however, it became clear that objective 2 and the related discussion confused all of us. We did not understand i) why the "historical belowground ecosystem" is important

Reviewer 1 also brought this to our attention. This sentence needs to be reworded to be more explicit for a broader audience. The 'historical belowground ecosystem' is the prairie systems under which the soils formed.

and ii) how objective 2 relates to the presented results. The question if these soils will continue to change is rhetoric (soil always continue to develop) and very unspecific.

Yes, this part of the sentence will also be reworded. We did not mean 'if', but 'how' soils will continue to change under annual cropping systems compared to reconstructed perennial systems.

The corresponding discussion section (4.3) is very short and speculative. Are root "turnover" (with a lifetime of a few years) and soil organic carbon turnover (decades to centuries) somehow related?

They are definitely related. Where does soil organic carbon come from? As roots cease to be roots, they turn into $CO_2$ or organic matter, influencing soil organic carbon turnover on the scale of days to centuries, even thousands of years for soils that have been occupied by roots for thousands of years.

Probably they are not. Coincidence is not causation. We wondered if objective 2 and section 4.3 are needed at all. If you think they are, please elaborate this part of the manuscript.

Yes, we will certainly elaborate on this.

Detailed comments: We did not fully understand the link between C:N and root depth.

Do you mean that C:N ratios increase with depth depending on species or on individual plants?

As we move deeper into the soil, root C:N ratios increase for all plants in this study.

P. 3, line 26: what does "sampling by replicate block from 31 October-25 November 2008" mean? Did you sample repeatedly? Explanation of "replicate block"-approach needed.

Yes, this is not clear and we will add more details.

P. 4, line 11/12: is there a difference between root measurements and root data?

Yes, there is. We should not have used them interchangeably here. Thanks for catching that.

P. 4, line 24/25: why different storage? Further explanation desired.

We explain that storage was different because soil from the first year was part of an incubation experiment.

P. 5, line 28: why the period between April 1st and November 30th to calculate the average root mass accumulation? Are these official dates? Further explanation needed.

These dates are the approximate window for plant growth in our region. We will include that information in the manuscript.

Table p. 10: unclear -> explanation of upper/lower case letters and meaning of those letters is missing; it could be part of the description of the table

This information is in the caption. Perhaps it did not come through with the format you were looking at.

Explanation pro glimmix on page 5/14 but not in table description.

Thank you, but not sure what you are looking for here.
P. 13, line 3: increase in root pool C:N ratio has not been reported previously in the literature: We would appreciate some information about previous research which focused on a related topic

We literally could not find any literature related to root pool C:N ratio.

P. 14, line 3/4: the pattern of distribution of what? Do you mean the vertical distribution of roots? What is place in this context?

Pattern of distribution of roots. We will add this to the text.

P. 14, line 30-32: For us, this sentence is very long and difficult to understand. No significant changes in soil C (changes in quantity or stocks?) at any depth but differences in quantity?

We will split this sentence up into segments that are easier to understand. No changes in soil C despite changes in root quantity.

"implementation of annual cropping systems": Do you refer to line 19? Experimental location was a site of cultivation under annual crops for over 100 years.

These soils were dominated by prairies for 10,000 years, so the shift to annual cropping systems 100 years ago is still relatively recent in terms of soil development. This information is important for us to add to the text.

Remarks concerning formal structures (typos, figures etc.):

P. 3, line 2: typo: 11 mg kg-1

Thanks.

Figure p. 6: why not making a title with total C, root C maize, etc. instead of letters a,b,c -> would be more clear and consistent with the following figures

Yes, we can change this.

Figures p.8 + 9: legend can be improved -> no units -> unclear & colors are not suitable

for black/white printing

We will add units to the legend and we used to different linetypes for different treatments in black and white printing, but see now that is not obvious.

P. 13, line 36: typo: this

Thanks.

---

## Author Comment (AC3) · 19 May 2017

Dietzel et al. report on a root study conducted at a field experiment where continuous corn is compared to reconstructed fertilized and unfertilized prairie stands. They have measured: 1) root profiles to depth of 1 m at the end of the growth season for six consecutive years, 2) root production (by regrowth cores) for 2 growing seasons to 30 cm depth, 3) root and soil C and N concentrations to 1 m depth. Extracting root for multiple growing seasons, multiple soils layers and multiple replicated treatments is by no mean easy, and the soil science community can certainly benefit from such precious

data. The authors report interesting findings: 1) the C/N ratio of root material increases with depth, which has potential implications for soil C storage, 2) the maize root profile is more uniform with depth than that of prairie species (confirmative), 3) fertilization of the reconstructed prairie greatly decreases root biomass.

However, I have some significant concerns with the study:

1) The continuous accumulation of maize roots throughout the 6-year period is quite troubling (Fig A3). The authors provide one reference (Dupont 2014) stating that intact prairie root (not maize) can be found in soil several years after cultivation. However, they ignore the substantial literature on maize roots that clearly indicates that maize roots decompose rapidly in soils, starting with the classical study of Mengel and Barber in 1974 (Agron. J. 66: 341-344) and several studies that have followed. Actually, Mengel and Barber (1974) state that root length and fresh weight decrease rapidly after maize has reached the reproductive stage. Here, Dietzel et al. themselves state in the abstract about maize roots that they are "non-structural-tissue dominated root C pool with fast turnover". They also indicate that the site was apparently under maize soybean rotation prior to starting the experiment, so why was there no accumulated maize root biomass at the start of the experiment (if the root accumulation theory is correct)? Unfortunately, in the present study the roots were not sampled at the same time each year (from early October to early November), and the accumulation of maize roots the last two years also corresponds to the 2 earliest sampling. A possible explanation is that the roots actually decomposed quickly in the field and that by sampling a month earlier by the end of the six-year period a greater number of non-decomposed roots were retrieved. Effects of inter-annual climatic variability on root growth is another potentially contributing effect. There are three implications from this: 1) apparent maize-root accumulation in the field over 6 years is probably an artefact, 2) the pool and rate modelling of Fig 3 and 4 is not justified (it did not bring much to the paper anyway), 3) the paper should have included a

much more throughout review of the literature about maize-root dynamics in field soils.

Thank you very much for bringing this to our attention. The possibility of such as artifact led us to additional analysis of our data. During this analysis, we found that the dates included in the methods section were not correct. We are very sorry for this mistake. Please find below Fig. 1 that illustrates that time of sampling was most likely not a contributing factor in the accumulation of root mass at any depth.

You bring up some other useful points here, thank you. We can certainly include more papers on the decomposition of maize roots. Our values for maize biomass taken after maize harvest are typically 10 percent of maize root values taken at maturity. This is in line with studies that have found that maize roots decompose rapidly. However, decomposition rate is an exponential function, with rapid decomposition occurring early and a slower rate of decomposition occurring later. Several months after maize maturity we are most likely sampling during a period of slower root decomposition.

We also questioned the lack of accumulated root mass from years previous to our experiment. We were careful to collect maize root samples 20 cm from our maize row and to plant our rows within 2-3 cm of the previous year's row. We assumed that the cropping legacy from years past would have left a more homogenous distribution of roots than the one we implemented.

2) The maize-root C profile is presented 3 times in the paper: 1) Fig 1 b, 2) Fig 2, and

3) Fig. A3. The 3 figures are in the same units (Mg ha-1), but I could not reconcile the data between them. The 2013 data of fig. 1 b (with largest root accumulation in the top soil) do not seem to correspond to the 2013 data of figure A3, which seems to show highest maize-root biomass in the deeper soil.

[Figure]

You are correct in not being able to reconcile the figures in the paper (1b and 2) with the figures in the appendix (A3) because the depth increments are not the same. Figure A3 shows the actual data, mass collected at 0-5, 5-15, 15-30, 30-60, and 60-100 cm depths. For example, the deepest depth increment is 40 cm long, resulting in high values relative to 5-15 cm, only 10 cm long. We used the values taken in these depth increments to break distribution into 5 cm increments, as described in P6, L 3-4. This gave a more accurate depiction of the distribution of both roots and organic C through the soil profile, shown in Figs. 1 and 2.

In addition, the maize root profile appears more even in Fig.2 than in Fig. 1 b, while it should be exactly the opposite (e.g. the 5-10 cm should have about half of the 0-5 cm in fig 2, if extrapolated from fig 1). Or are the two figures exactly the same? But why figure 2 then? The 3 figures should have been reconciled and presented as one main figure in logical units, and then the results compared to the literature.

Fig. 1b and Fig.2 are drawn from the same dataframe. Fig. 1 shows distribution patterns that, as you point out, are not visible in Fig. 2, especially for maize. We use Fig. 1 to discuss root and soil C distribution in the soil profile. Figure 2 shows absolute differences in root pool mass among treatments, not visible in Fig. 1. Fig. 2 is used to point out and discuss these basic differences.

3) Data from the root regrowth cores are not clearly presented, but used for a direct extrapolation of a root turnover rate in the top 30 cm. Summary data (without statistics) are presented in g m-2 in Table 4, making it difficult to compare to the root pool data presented in Mg ha-1. The maize root productivity appears low and I am missing a coherent evaluation of the C input in the context of published studies.

Sorry these data are not clear, we will improve this in the paper as well as change the units to correspond with other reported units. The maize root productivity is low

because samples were taken 2-3 months after maize maturity, which we will also high-light.

4) The implications for C storage presented in this paper are largely hypothetical and somewhat contradictory. The present paper contains no significant result to link root biomass profiles to soil C profiles. While it is OK to briefly elaborate about possible implication of a higher root C/N ratio with depth, this should not be the main part of the discussion, which should instead focus on actual significant results.

The implications for C storage presented in this paper are partly hypothetical, but are complex and novel enough to warrant extensive discussion. The mystery of the disproportionally large stock of SOM relative to root C input found in deep soil has been unsolved since first noticed over 100 years ago and root C:N ratio increase with depth is a new and useful piece of information. We do not have direct evidence to link root biomass profiles to soil C profiles, but what we do have, combined with what we are able to model, is better than anything that has been previously published.

In addition, I could not reconcile the two ideas presented here about the effect of root C/N ratio on C storage in soils. On the hand, the authors argue that a lower C/N ratio for maize root favours C storage in soil as compared to prairie roots (p14, line 11-12). On the other hand, they also argue that an increasing C/N ratio of roots with depth in the soil profile also favours C storage (e.g. p1, line 10-12). The potential attempt to reconcile these two contradictory effects of root C/N ratio on soil C was unconvincing (p 14).

We will reword P1, line 10-12 - "In all treatments we found that root C:N ratios increased with depth, which may help explain why an unexpectedly large proportion of soil organic C is found below 20 cm." It is meant to suggest that increaseing C:N ratios with depth play a role in an unexpectedly large proportion of soil organic C found below 20 cm, not

[Figure]

that the increase in C:N ratio directly contributes C storage. The relationship between C:N ratios and C storage is not necessarily intuitive, as we later describe in the text.

In conclusion, Dietzel et al. have collected an impressive data set on maize and prairie roots following maize-soybean rotation. The dataset appears to suffer from some artefacts, but root studies are difficult and shortcomings could have been acknowledged.

Thank you.

The data themselves are neither clearly presented nor sufficiently discussed in light of the literature. A main finding is largely ignored, i.e. the dramatic reduction of root biomass by fertilization in prairie systems.

Indeed, apart from the figures, we did include only three sentences on this finding. The effect of N fertilization on perennial root biomass is well-known (Troughton 1960, Thornley 1972, Gregory 2007) and we did not consider this plant response to be particularly important to a soils audience.

By contrast, the authors focus on an uncertain modelling and non-verifiable considerations about the effect of root C/N ratio on soil C. A focus on significant results and discussion of these results in light of the literature would have better served this study

While modelling is always uncertain, the fits of the models we used were very good and we stand behind the conclusions drawn from this effort. The effect of root C:N ratio on soil C may never be verifiable because it is a centuries-long process that is difficult to measure, but root C:N ratio plays some role in the development of the soil C profile. This manuscript is the very first to question this role and work with existing data to propose a mechanism by which root C:N ratio contributes to soil C profile development. It is our hope that by focusing our paper on root C:N ratios and current soil C profiles, we are inspiring and supporting future studies that will examine this important relationship more closely. Merely commenting on this possible relationship in a manuscript focused on a comparison of maize and prairie roots would not reach the intended audience or

provide direction for related experiments.

Gregory, P.: Plants Roots: Growth, Activity and Interaction with Soils, in Plant, Roots and the Soil, pp. 5-7, Blackwell Publishing Ltd., 2007.

Thornley, J. H. M.: A balanced quantitative model for root: shoot ratios in vegetative plants, Annals of Botany, 36(145), 431-441, 1972.

Troughton, A.: Further studies on the relationship between shoot and root systems of grasses, Journal of the British Grassland Society 15, 41-47, 1960.

[Figure]

**Fig. 1.** Root pool mass by sampling time for each depth. 5 is 0-5 cm, 15 is 0-15 cm, 30 is 15-30 cm, 60 is 30-60 cm, and 100 is 60-100 cm.